evolution

sperm chemotaxis, post-copulatory sexual selection, major histocompatibility complex, sperm competition, *in vitro* fertilization

**Author for correspondence:**
John L. Fitzpatrick
e-mail: john.fitzpatrick@zoologi.su.se

# Chemical signals from eggs facilitate cryptic female choice in humans

John L. Fitzpatrick[1,2], Charlotte Willis[3], Alessandro Devigili[1], Amy Young[2], Michael Carroll[4], Helen R. Hunter[5] and Daniel R. Brison[3,5]

[1]Department of Zoology, Stockholm University, Svante Arrhenius väg 18B, 10691 Stockholm, Sweden
[2]Faculty of Biology, Medicine and Health, University of Manchester, Oxford Road, Manchester M13 9PT, UK
[3]Maternal and Fetal Health Research Centre, Faculty of Biology, Medicine and Health, University of Manchester, Manchester Academic Health Sciences Centre, Manchester, UK
[4]Department of Life Sciences, Faculty of Science and Engineering, Manchester Metropolitan University, Manchester M1 5GD, UK
[5]Department of Reproductive Medicine, Saint Mary's Hospital, Manchester University NHS Foundation Trust, Manchester Academic Health Sciences Centre, Oxford Road, Manchester M13 9WL, UK

JLF, 0000-0002-2834-4409; AD, 0000-0001-8104-5195

Mate choice can continue after mating via chemical communication between the female reproductive system and sperm. While there is a growing appreciation that females can bias sperm use and paternity by exerting cryptic female choice for preferred males, we know surprisingly little about the mechanisms underlying these post-mating choices. In particular, whether chemical signals released from eggs (chemoattractants) allow females to exert cryptic female choice to favour sperm from specific males remains an open question, particularly in species (including humans) where adults exercise pre-mating mate choice. Here, we adapt a classic dichotomous mate choice assay to the microscopic scale to assess gamete-mediated mate choice in humans. We examined how sperm respond to follicular fluid, a source of human sperm chemoattractants, from either their partner or a non-partner female when experiencing a simultaneous or non-simultaneous choice between follicular fluids. We report robust evidence under these two distinct experimental conditions that follicular fluid from different females consistently and differentially attracts sperm from specific males. This chemoattractant-moderated choice of sperm offers eggs an avenue to exercise independent mate preference. Indeed, gamete-mediated mate choice did not reinforce pre-mating human mate choice decisions. Our results demonstrate that chemoattractants facilitate gamete-mediated mate choice in humans, which offers females the opportunity to exert cryptic female choice for sperm from specific males.

# 1. Introduction

Prior to mating, animals advertise and evaluate an array of often conspicuous visual, acoustic and chemical sexual signals [1,2]. These pre-mating sexual signals offer the choosing sex (typically females) the opportunity to assess the quality and genetic compatibility of potential mates [1–4]. After mating, communication between the sexes continues but is restricted to chemosensory communication between gametes and, in the case of internal fertilizers, the female reproductive tract [5]. Such post-mating chemosensory communication between eggs and sperm can facilitate gamete-mediated mate choice, allowing eggs to exert cryptic female choice and bias fertilizations towards specific males [6,7]. However, our understanding of the potential for gamete-mediated mate choice remains limited in internally fertilizing species, and is completely unexplored in humans.

Sperm chemoattraction, a remote form of chemical communication between eggs and sperm occurring before gamete contact, is a widespread mechanism for increasing sperm density around unfertilized eggs in animals [5]. In broadcast spawning marine invertebrates, where adults are unable to express pre-ejaculatory mate choice, chemoattractants increase fertilization rates by increasing the effective target size of the egg, maintain species barriers by preferentially recruiting conspecific sperm, and allow sperm and eggs to exercise gamete-mediated mate choice [6–9]. Chemoattractants in marine invertebrates can also preferentially recruit sperm from specific, presumably more compatible males by remotely altering sperm swimming physiology and behaviour, thereby increasing fertilization rates, embryo viability and offspring survival [7]. In internally fertilizing species, females can exercise cryptic female choice through interactions between sperm and the female reproductive tract, influencing the number of sperm a female retains and/or sperm swimming performance [10,11]. Subsequent interactions between eggs and sperm can also facilitate gamete-mediated mate choice in internal fertilizers. For example, in house mice (*Mus domesticus*) Firman & Simmons [12] found that eggs were preferentially fertilized by sperm from less related males during *in vitro* fertilizations (IVF), and suggested that either direct interactions among cell-surface proteins on gametes or differential responses to chemoattractants could explain these effects. In mammalian reproduction, chemoattraction is the last of a series of sperm guidance mechanisms (including positive rheotaxis and thermotaxis) that acts to recruit capacitated sperm to eggs [5,13]. By contrast with marine invertebrates, mammalian sperm lack species-specificity in responses to chemoattractants [14], suggesting that pre-mating species recognition mechanisms may reduce the need for post-mating processes to reinforce species barriers. Nevertheless, mammalian chemoattractants could play a post-mating role in gamete-mediated mate choice, either to maximize genomic compatibility between potential mates [3,4] or to reinforce or override pre-mating mate choice decisions [11,15]. However, the potential for chemoattractants to serve a sexually selected role in human reproduction remains unexplored.

Here, we assess if follicular fluid, a source of sperm chemoattractants [16], differentially regulates sperm behaviour to reinforce pre-mating mate choice decisions and mediate fertilization success in humans. Human sperm respond to chemoattractants present in the follicular fluid surrounding eggs (most likely progesterone [5], although this remains a source of ongoing debate) by altering their swimming behaviour to orient towards, and accumulate in, follicular fluid [16]. Sperm behavioural responses can differ among follicular fluids, such that follicular fluids from different females exhibit variation in their ability to attract sperm from the same male [16]. Moreover, females producing follicular fluid that was better at causing an accumulation response in sperm also produce eggs that achieved higher fertilization rates in clinical IVF cycles [16]. Thus, differential responses in sperm behaviour to follicular fluid have the potential to facilitate gamete-mediated mate choice in humans. We investigated this potential using two distinct experimental designs, exposing sperm to follicular fluid from two females either simultaneously or non-simultaneously, and report robust evidence that sperm accumulation is influenced by the interactive effects between males and females.

# 2. Material and methods

## (a) Sample acquisition and clinical data

We obtained follicular fluid and sperm samples from couples undergoing assisted reproductive treatment (IVF; intracytoplasmic sperm injection, ICSI) at St Mary's Hospital, Manchester, UK, with written informed patient consent and approval from Central Manchester Research Ethics Committee (electronic supplementary material). We specifically focused on couples receiving assisted reproductive treatment, rather than, for example, performing assays using sperm from males not seeking fertility treatment, as one of our aims was to investigate the link between partner choice and gametic interactions. Samples were obtained using standard clinical practices [17] (see electronic supplementary material). All data were collected blind to the treatments and patient identity to ensure that patient confidentiality was maintained to comply with the WHO Good Clinical Research Practice guidelines [18] and the Human Fertilization and Embryology Authority Code of Practice [19], and also ensure that the researcher was blinded from identifying which samples originated from each couple during the experiment.

Information relating to the participant's fertility procedure was collected, including the type of fertilization method used (IVF or ICSI), number of oocytes retrieved, number of oocytes successfully fertilized, embryo quality score, pregnancy outcome and live birth success. For couples undergoing IVF, fertilization success was calculated as the number of fertilized embryos divided by the number of oocytes retrieved and inseminated, while for couples undergoing ICSI, fertilization success was calculated as the number of fertilized embryos divided by the number of oocytes injected. Embryo quality was determined using an embryology morphology grading scheme (see electronic supplementary material). Pregnancy outcome was scored based on evidence of implanted embryos, while live births were treated as successful outcomes relative to all other outcomes (see electronic supplementary material). Ejaculate traits differ between patients being treated with IVF or ICSI; for example, sperm density is lower in ICSI than IVF patients in the non-simultaneous choice experiment (see below) (linear mixed model, LMM: $\chi^2 = 5.2$, $p = 0.02$). Therefore, we were cautious in how we analysed data from these two treatment groups. However, our experimental design excluded males with severe male factor infertility, as such males would not have sufficient sperm to be used in the experimental assays (i.e. males undergoing ICSI were diagnosed with either male factor subfertility or were normospermic men who suffered poor fertilization with IVF in a previous cycle). Data analyses examined the impact of including ICSI patients in statistical models either by removing these patients or including the type of fertility treatment as a fixed effect in the model (described below).

## (b) Experimental overview

To test if follicular fluid influences sperm behaviour, we adapted a classic dichotomous mate choice assay to the microscopic scale (i.e. simultaneous presentation of two stimuli; figure 1). We performed two experiments using a North Carolina II cross-classified block design [20]. This experimental design facilitated the examination of female, male and female–male interacting effects on sperm behaviour in follicular fluid. Each experimental block comprised the follicular fluid and sperm samples from a unique set of two couples, exposing sperm from each male to follicular fluid from their partner and a non-partner (figure 1a,b). We performed two cross-classified experiments that differed in how sperm experienced the choice of follicular fluid from different females, being either 'simultaneous' or 'non-simultaneous' (figure 1c,d). Sperm responsiveness to follicular fluid was quantified by counting the number of sperm accumulating in the follicular fluid from each

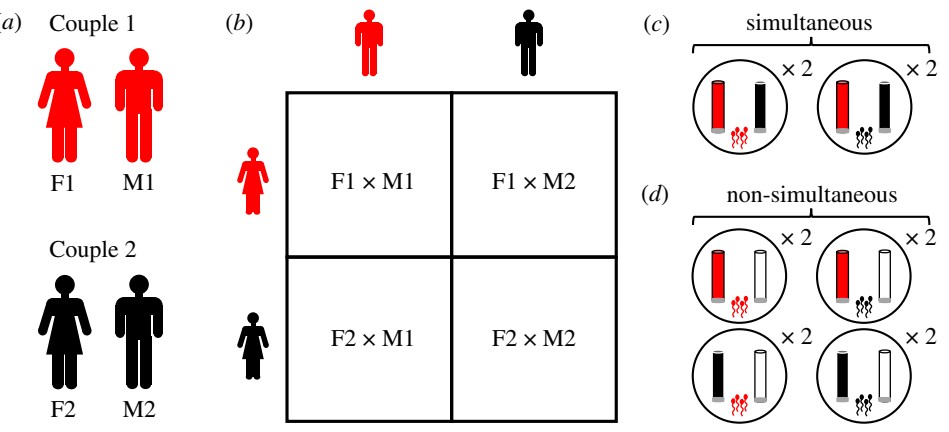

**Figure 1.** Microscopic mate choice. An overview of the experimental design used to assess variation in sperm accumulation responses to follicular fluid from different females. (*a*) Each experimental block consisted of samples of follicular fluid and sperm that were obtained from two couples—couple 1, comprising female 1 (F1) and male 1 (M1), and couple 2, comprising female 2 (F2) and male 2(M2)—undergoing clinical assisted reproductive treatment. (*b*) Sperm accumulation in follicular fluid was assessed by crossing females and males in all possible combinations for each experimental block in the cross-classified design. Each cross was replicated twice for every female–male combination. Thus, in each experimental block, sperm were exposed to follicular fluid from both their partner (from the same couple) or a non-partner (from a different couple). An example of a single block in the experimental design is presented, where sperm were exposure to follicular fluid (housed in microcapillary tubes) from two females under (*c*) *simultaneous* (*n* = 8 blocks) or (*d*) *non-simultaneous* (*n* = 22 blocks) experimental conditions. The microcapillary tubes were sealed with a plug (indicated in grey) at the end of the tube where sperm were added to the Petri dish. Thus, to enter the microcapillary tube, sperm had to swim the length of the microcapillary tube. In the simultaneous experimental design sperm were presented with a simultaneous choice of follicular fluid from two females, while in the non-simultaneous experimental design sperm were presented with a choice of follicular fluid from one female (either the partner or non-partner) and a control medium. The number of sperm that successfully entered the microcapillary tube was counted by light microscopy at 300× magnification to quantify sperm accumulation in and responsiveness to the follicular fluid. (Online version in colour.)

female. Each assay was replicated twice and repeatability among experimental replicates was high (simultaneous choice experiment: $R = 0.96$, 95% CI = 0.92–0.98, $p < 0.001$; non-simultaneous choice experiment: $R = 0.97$, 95% CI = 0.95–0.98, $p < 0.001$, see electronic supplementary material). High repeatability is consistent with a chemotactic response rather than mechanical trapping effects, where sperm accumulate due to adsorption to the experimental apparatus, which is unlikely to exhibit high repeatability between experimental replicates [21].

## (c) Simultaneous choice of follicular fluid experiment

The simultaneous choice experiment consisted of 16 couples, comprising eight blocks of factorial crosses (14 IVF and 2 ICSI treatments; note that excluding ICSI patients from the analyses did not qualitatively alter our findings; see below). In each block, sperm were presented simultaneously with follicular fluid in 2 µl microcapillary tubes in a Petri dish from two females (a partner and a non-partner) to determine if sperm preferentially and consistently swim towards and accumulate in the follicular fluid of a specific female (figure 1*c*; electronic supplementary material). Thus, sperm had to swim the length of the microcapillary tube (approx. 30 mm), moving up the chemoattractant gradient, to enter the microcapillary tube containing the follicular fluid. Following sperm addition, Petri dishes were left undisturbed in the incubator for 1 h to allow the sperm time to migrate towards and accumulate within the microcapillary tubes.

## (d) Non-simultaneous choice of follicular fluid experiment

We evaluated sperm responses to follicular fluid when exposure to follicular fluid was non-simultaneous (figure 1*d*): sperm were presented with the choice between the follicular fluid from one female (either the partner or non-partner) and a control sperm preparation medium (SpermRinse) solution ($n = 44$ couples, 22 blocks of factorial crosses; 30 IVF and 14 ICSI treatments). This experimental design allowed us to test if sperm preferentially and consistently accumulate in the follicular fluid of a

specific female relative to a control solution. Assays were performed in Petri dishes primarily as described in the 'Simultaneous choice of follicular fluid experiment' section (see electronic supplementary material for details of the minor modifications in the experimental design), although in the case of the non-simultaneous choice experiment only one female's follicular fluid was presented in each Petri dish (figure 1*d*).

## (e) Quantifying sperm accumulation in follicular fluid

One hour after sperm addition, the microcapillary tubes were removed from the Petri dish and placed on a microscope slide (Menzel-Glaser, Braunschweig, Germany). The number of sperm present in the microcapillary tubes were counted under 300× magnification using an Olympus IMT2 inverted microscope. Because microcapillary tubes are three-dimensional objects, we adjusted the plane of focus as the field of view moved along the length of the microcapillary tube to ensure that sperm were counted on all focal planes of the tube.

## (f) Follicular fluid and sperm swimming behaviour

Variance in sperm responses to follicular fluid can be caused by differential sperm chemotactic responses and/or differential responses in sperm swimming speed (i.e. chemokinesis). To evaluate the potential role of chemokinesis in mediating sperm responses to follicular fluid we characterized sperm performance in 12 males from a subset of six experimental blocks from the non-simultaneous choice experiment. Sperm swimming characteristics were quantified using computer-assisted sperm analyses (CASA). Sperm swimming characteristics from each male were assessed under three different treatments: in follicular fluid from their partner, follicular fluid from a non-partner, and in sperm preparation medium (SpermRinse) as a control (see electronic supplementary material).

## (g) Statistical analyses

To investigate whether sperm respond differentially to follicular fluid we performed a series of sequential two-way analysis of variance (ANOVA) models to estimate the female, male and

female–male interaction effects on sperm accumulation in the simultaneous ($n = 8$ blocks) and non-simultaneous ($n = 22$ blocks) choice experiments. Analyses were performed separately for each experimental block of factorial crosses and then combined in Microsoft Excel (v.16.16.13) into a final model using a 'North Carolina II' block design [20]. In the simultaneous choice experiment, we treated *sperm accumulation* (i.e. the number of sperm counted) in the microcapillary tube housing the follicular fluid as the response variable. In the non-simultaneous choice experiment, sperm accumulation was almost 10 times greater in follicular fluid ($435.0 \pm 39.0$) than the control solution ($45.1 \pm 3.0$, generalized linear mixed model (GLMM): fixed intercept: $Z = 12.2$, $p < 0.001$), confirming the chemoattractant properties of follicular fluid. However, as our aim was to assess how sperm responds to follicular fluid of different females, we treated *sperm responsiveness* to follicular fluid, quantified as the difference in the number of sperm accumulating in the microcapillary tube containing the follicular fluid relative to the microcapillary tube containing the control solution, as the response variable in the non-simultaneous choice experiment. In both the simultaneous and non-simultaneous experiments, we avoided interpreting significant female or male main effects in the presence of a significant female–male interaction. When blocks containing ICSI patients were removed from the analyses we obtained qualitatively similar results (see electronic supplementary material, table S1) and therefore we present results from the full dataset.

We next examined if sperm accumulation and responsiveness are influenced by the origin of the follicular fluid (i.e. partner versus non-partner follicular fluid). In the simultaneous choice experiment, we fitted a GLMM with a logit link function, treating sperm accumulation (i.e. the number of sperm counted) in either the partner or non-partner follicular fluid, which represented a binary choice of simultaneously presented follicular fluid (figure 1c), as a binomial response variable. The simultaneous choice model was fitted with fertility treatment (IVF versus ICSI) as fixed effects and the female, male, and female–male interaction, the experimental block and observation number (to account for overdispersion) as random effects. In the non-simultaneous choice experiment, we fitted a LMM treating sperm responsiveness (log10 transformation on positivized values) as the response variable, with follicular fluid origin (partner versus non-partner) and fertility treatment (IVF versus ICSI) as fixed effects (note the non-significant interaction term between the categorical fixed effects was dropped from the model and sperm density was removed from the final model as inclusion of this variable impaired model fit), and female, male, female–male interactions, and experimental block as random effects. Note that we obtained qualitatively similar results when we assessed if sperm accumulation and responsiveness were influenced by the origin of follicular fluid using simplified models where the number of random effects present in our main analyses were reduced (see electronic supplementary material).

We then explored if sperm swimming behaviour was influenced by follicular fluid (compared to a control solution) and if sperm behaviour differs when swimming in follicular fluid from a partner compared to a non-partner. We used principal components (PC) analysis as a data reduction method to reduce the seven highly correlated sperm swimming parameters produced by CASA into two PC's with eigenvalues greater than one (electronic supplementary material, table S2). To assess if sperm swimming speed differs when swimming in follicular fluid compared to a control solution, we fitted a LMM with experimental medium (follicular fluid versus a control solution), fertility treatment (IVF versus ICSI), and their interaction as fixed effects and male identity and experimental block as random effects. Next, we used a LMM to examine if sperm swimming speed is influenced by the origin of the follicular fluid (partner versus non-partner), treating follicular fluid origin, fertility treatment (IVF versus ICSI) and their interaction as fixed effects, while including

male identity and experimental block as random effects. To assess if sperm responsiveness to follicular fluid was influenced by variance in sperm swimming speed among males, we fitted a LMM with sperm responsiveness (sperm in follicular fluid—sperm in control) as the response variable, with sperm swimming speed (PC1 and PC2), the density of sperm added to the Petri dish (which is positively related with sperm accumulation in the microcapillary tubes, LMM: $\chi^2 = 15.2$, $p < 0.001$) and fertility treatment (IVF versus ICSI) as fixed predictor variables and male identity and experimental block as random effects.

Finally, we examined whether follicular fluid that preferentially attract sperm from their partner (relative to a non-partner) had higher fertilization/embryo quality/pregnancy/live birth success during IVF treatment than follicular fluid less capable of attracting sperm from their partner. All sperm data were derived from replicate mean values (see electronic supplementary material). We treated the proportion of eggs fertilized, pregnancies success (0 or 1) and live birth success (0 or 1) as binomial response variables and fitted GLMMs with a logit function. In the simultaneous choice experiment, *partner sperm preference* (i.e. the difference in sperm accumulation to the partner versus non-partner follicular fluid) was treated as a continuous fixed effect, and the experimental block and observation number included as random effects. In the non-simultaneous choice experiment, *partner sperm responsiveness* (i.e. partner follicular fluid—control sperm count) was treated as a continuous fixed effect, and the experimental block and observation number included as random effects. For embryo quality, we used LMMs with partner sperm preference or partner sperm responsiveness as predictor variables for the simultaneous and non-simultaneous choice experiments, respectively, and fitted models with experimental block as a random effect. Analyses of these fitness variables excluded patients treated using ICSI, as fertilizations using ICSI involve sperm being injected into the egg rather than sperm-directed movement towards the egg.

All analyses were performed in R Studio v.1.1.463 [22], with LMM and GLMM models fitted using the *lme4* package [23]. In GLMM and LMM models, parameters were estimated using the Laplace approximation of log-likelihoods and the Satterthwaite's method, respectively. GLMMs were initially fit with the bobyqa optimizer, but in cases where model convergence failed we set the nAGQ scalar to zero. Model diagnostics were performed by assessing overdispersion using the *RVAideMemoire* package in R [24] and testing for uniform distribution of the scaled residuals in the *DHARMa* package in R [25].

## 3. Results

When sperm were presented with a simultaneous choice of swimming towards follicular fluids from two females (a partner and a non-partner, $n = 16$ couples, eight blocks of factorial crosses; figure 1c), sperm accumulation in follicular fluid was significantly influenced by the interactive effect of female–male identity ($F_{8,32} = 19.38$, $p < 0.001$; figure 2a, table 1a). However, in internally fertilizing species such as humans, sperm are never presented with the simultaneous choice of follicular fluid from more than one female. Therefore, we performed a second cross-classified experiment under biologically relevant conditions, where sperm were non-simultaneously exposed to follicular fluid from two females ($n = 44$ couples, 22 blocks of factorial crosses; figure 1d). In the non-simultaneous choice experiment, sperm were given the choice between the follicular fluid from one female (either the partner or non-partner) and a control solution (sperm preparation medium). When sperm were presented with the non-simultaneous choice of follicular fluid, sperm responsiveness was also influenced by the

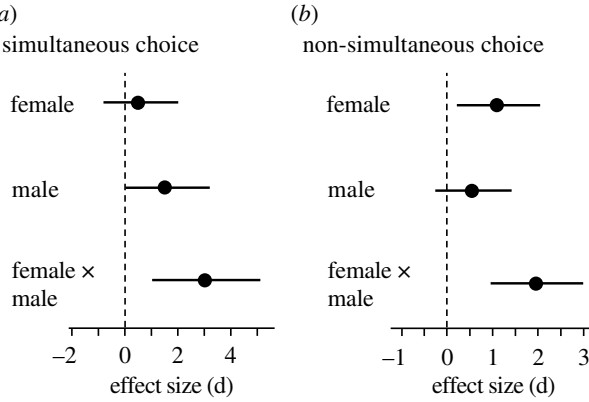

(a) simultaneous choice

(b) non-simultaneous choice

**Figure 2.** The effect of female, male and female–male interactive effects on sperm accumulation in the (a) simultaneous and (b) non-simultaneous choice experiments. The effect size (Cohen's d) and 95% confidence intervals are presented for each effect in the cross-classified design. Plots are for illustrative purposes only, as values were derived from standard $F$-test effect size calculations. Effects are considered significant when confidence intervals do not overlap with zero.

interaction between female and male identity ($F_{22,88} = 21.82$, $p < 0.001$; figure 2b, table 1b). The significant interactive effects of female–male identity on sperm behaviour remained when we examined IVF and ICSI patients separately in the simultaneous and non-simultaneous choice experiments (electronic supplementary material, table S1).

To evaluate the potential role of partner effects, we assessed if the female–male interactive effects in sperm accumulation/responsiveness are influence by the origin of follicular fluid. In the simultaneous choice experiment, sperm accumulation did not differ between follicular fluid of the partner ($515.7 \pm 79.9$, mean $\pm$ s.e.) or non-partner ($441.9 \pm 51.0$, GLMM; fixed intercept: $Z = -0.29$, $p = 0.77$; fertility treatment effect (IVF versus ICSI): $Z = 0.71$, $p = 0.48$), indicating that sperm do not preferentially accumulate in the follicular fluid of their partner. Similarly, when we assessed sperm responsiveness to follicular fluid in the non-simultaneous choice experiment, patterns of sperm responsiveness were not affected by the origin of the follicular fluid (follicular fluid origin: $\chi^2 = 0.32$, $p = 0.57$; fertility treatment: $\chi^2 = 3.23$, $p = 0.07$).

Although follicular fluid influenced sperm behaviour in the patients we considered (electronic supplementary material, table S3), sperm swimming behaviour did not differ when exposed to follicular fluid from either a male's partner or a non-partner (electronic supplementary material, table S3). Moreover, sperm responsiveness to follicular fluid was not related to sperm swimming speed, suggesting that patterns of sperm accumulation are not explained by sperm chemokinetic responses (sperm velocity PC1: $\chi^2 = 0.25$, $p = 0.61$; sperm velocity PC2: $\chi^2 = 0.01$, $p = 0.94$; fertility treatment: $\chi^2 = 0.03$, $p = 0.87$; sperm density: $\chi^2 = 1.42$, $p = 0.23$).

We found limited evidence that fitness measures were influenced by sperm responses to follicular fluids. In the simultaneous choice experiment, fertilization rates using IVF were higher when sperm were more responsive to their partner's follicular fluid (i.e. partner sperm preference was stronger, $Z = 2.25$, $p = 0.02$, electronic supplementary material, figure S1a). Similarly, there was a statistical trend suggesting that embryo quality increased when partner sperm preference was stronger ($\chi^2 = 3.51$, $p = 0.06$). However, these relationships were driven entirely by two outlying data points (see electronic supplemental material, figure S1a). Partner sperm preference

did not predict whether IVF treatment resulted in clinical pregnancy ($Z = 0.82$, $p = 0.40$) or live births ($Z = 1.11$, $p = 0.27$). In the non-simultaneous choice experiment, partner sperm responsiveness did not predict the proportion of eggs fertilized using IVF ($Z = 0.52$, $p = 0.60$, electronic supplementary material, figure S1b), embryo quality ($\chi^2 = 1.28$, $p = 0.26$), clinical pregnancy ($Z = -0.80$, $p = 0.42$) or live birth success ($Z = -1.34$, $p = 0.18$).

## 4. Discussion

Chemical communication between eggs and sperm is critical for fertilization. As sperm make their way towards eggs, sperm behaviour is influenced by signals released from unfertilized eggs and/or the female's reproductive tract, leading to the traditional view that chemical signals act only to guide sperm to eggs in internal fertilizers, like humans [5]. Our findings challenge this long-standing paradigm. We found concordant patterns of sperm responses to follicular fluid under two distinct experimental conditions, performed with two independent groups of couples that were temporally separated. Our results demonstrate that sperm accumulation in follicular fluid depends on the specific combinations of follicular fluid and sperm, and that follicular fluid preferentially attracts sperm from specific males. The non-random, repeatable sperm accumulation responses we detected suggest that chemical communication between eggs and sperm allows females to exert 'cryptic choice' over which sperm fertilize their eggs.

Female–male interactive effects during reproduction are a hallmark of cryptic female choice [10]. Such differential sperm responses to follicular fluid have the potential to influence fertilization success between specific female–male partners. Sperm number is a key determinant of fertilization success [26]. In humans, a minute fraction of ejaculated sperm makes its way up the fallopian tube to the site of fertilization (mean = ~250 sperm [27]). Of these few remaining sperm, roughly one in ten is capacitated (a biochemical process required for fertilization capacity) and capable of responding to chemoattractants and fertilizing the egg [28]. The ever-dwindling number of sperm capable of fertilizing eggs as they move through the female's reproductive tract suggest that the capacity for chemoattractants to differentially recruit sperm from specific males could make the difference in ensuring fertilization success.

Yet, despite the potential for differential chemotactic responses to influence fertility, we found only weak evidence that sperm responses to chemoattractants influence fertilization success and later fitness measures. This contrasts with findings in marine invertebrates, where differential sperm responses to chemoattractants can influence both fertilization success and subsequent embryo viability [6,29]. However, the lack of a clear relationship between partner sperm preference and/or partner sperm responsiveness and fitness measures during IVF is perhaps not surprising given the constraints of the clinical setting where our experiments took place. Clinical practices place sperm in close contact with eggs, potentially minimizing the importance of chemoattractants prior to fertilization, and attempt to maximize fertilization success by using sperm concentrations several orders of magnitude greater than are found in the fallopian tube at the site of fertilization *in vivo* (e.g. typically 20 000 sperm per egg, [17]). Downstream clinical treatment of embryos following fertilizations also removes

**Table 1.** Sources of variation in sperm accumulation in the (*a*) simultaneous and (*b*) non-simultaneous choice experiments. The degrees of freedom (d.f.) and the sum of squares (SS) were calculated individually for each experiment block using a series of sequential two-way ANOVAs. The d.f. and SS from all experiment blocks were summed and combined to estimate the mean squares (MS) for each analysis. The d.f. for each block was calculated by multiplying the number of females, the number of males and the number of replicate crosses minus one for each block. The dfs from main effects and error estimates were summed across blocks. *F*-values were obtained for male and female effects by dividing their respective MS values by the interaction MS. *F*-values for the interaction term were calculated by dividing the interaction MS value with the error MS. Statistically significant values are in bold. Due to differences in sperm number among males (but not between replicates for each male within an experimental block), we did not interpret male effects in our models, nor did we interpret main effects when significant interactive effects were detected.

| source of variation | d.f. | SS | MS | F | p |
|---|---|---|---|---|---|
| *(a) simultaneous choice experiment* | | | | | |
| female | 8 | 528385.1 | 66048.1 | 0.81 | 0.61 |
| male | 8 | 3362391.1 | 420298.9 | 5.15 | **0.02** |
| female × male interaction | 8 | 652583.1 | 81572.9 | 19.38 | **<0.001** |
| error | 32 | 134675.5 | 4208.6 | | |
| *(b) non-simultaneous choice experiment* | | | | | |
| female | 22 | 9358699.9 | 425395.4 | 7.05 | **<0.001** |
| male | 22 | 2412092.4 | 109640.6 | 1.82 | 0.09 |
| female × male interaction | 22 | 1328351.4 | 60379.6 | 21.82 | **<0.001** |
| error | 88 | 243507.5 | 2767.1 | | |

potentially important biological processes. Thus, while challenging to detect *in vitro*, follicular fluid-mediated differential recruitment of sperm could play an important role during *in vivo* fertilizations in humans, although this requires further validation. An important next step is to determine if incorporating considerations of chemical communication between gametes into clinical practices could improve not only fertilization success but the quality of developing embryos both prior to and post-implantation.

Female–male interactive effects are also characteristic of mate choice for genetic compatibility generally [3,4,6,7]. Thus, the female–male interactive effects we detected raise the possibility that preferential sperm accumulation reflects a chemoattractant-mediated mechanism occurring prior to direct sperm-egg interactions to avoid post-mating genomic incompatibilities (*sensu* 12). For example, sperm could swim preferentially towards the follicular fluid from their partner, provided human pairing reflect mate choice for genetic compatibility at the major histocompatibility complex (MHC), a diverse chromosomal region that functions in immune defence (although whether this is the case remains controversial, [30–34]). Under this scenario, sperm responses to their partner's (or indeed non-partner's) follicular fluid may reflect the degree of genetic compatibility between the pair. Alternatively, as the couples in our study were undergoing fertility treatment, the interactive effect between males and females could stem from a clinical pathology that impairs chemical communication between gametes. This has direct clinical relevance as a high proportion (32%) of couples undergoing fertility treatment in the UK have a cryptic ('unexplained' or idiopathic) cause to their infertility [35]. In cases of idiopathic infertility, sperm may swim preferentially towards follicular fluid from random (i.e. non-partner) females over follicular fluid from their partner. However, we find no support for either of these possibilities as female–male interactive effects were not explained by differential responses to either partners or non-partners. Nevertheless, considering female–male interactive effects when examining the mechanistic underpinnings of

chemical communication between gametes will help to clarify the factor(s) influencing sperm accumulation in follicular fluid.

Our results demonstrate that patterns of sperm accumulation are shaped by combinatory female–male effects and cannot be explained by differences in the quality and/or amount of chemoattractants present in the follicular fluid of different females or by differences in male ejaculate quality. Female–male interactive effects are a key diagnostic required for demonstrating cryptic female choice of sperm [12]. Thus, despite ample scope for humans to exercise pre-mating mate choice [2,30], chemosensory communication between gametes retains a role in selectively recruiting sperm. Indeed, in their initial demonstration that human sperm are attracted by chemical signals in follicular fluid more than 25 years ago, Ralt *et al*. [16] reported variation in sperm responsiveness to follicular fluids from different females. However, until now, the implications of this finding have not been explored. Our results imply that chemoattractants may allow females to exert post-mating (i.e. cryptic female choice) gamete-mediated mate choice. These findings extend the traditional view that chemoattraction solely plays a role in increasing sperm-egg interactions in humans [5] and instead suggest that chemical communication between eggs and sperm may also have a sexually selected role. A critical next step is to determine if such female–male interactive effects are a common feature of mammalian reproduction, including humans not undergoing assisted fertility treatments (although this is logistically and ethically challenging), and to examine the potential for gamete-mediated mate choice to influence embryo quality under biologically relevant conditions. Nevertheless, our findings suggest that chemosensory-driven interactive responses to chemoattractants probably span the animal tree of life and potentially provide a widespread mechanism of gamete-mediated mate choice that is currently underappreciated. Clarifying the significance of chemical communication between human gametes during fertilizations and uncovering the molecular mechanisms influencing differential sperm response may aid in the development of new approaches for diagnosing

and treating unexplained infertility and improving the efficiency and safety of assisted reproductive treatments.

Data accessibility. The dataset associated with this study is available from the Dryad Digital Repository: https://doi.org/10.5061/dryad.wdbrv15kb [36].

Authors' contributions. J.L.F. and D.R.B conceived the study, obtained funding and drafted the manuscript. C.W., A.D. and A.Y. performed the majority of the technical work in collecting the data with supervision from J.L.F., H.R.H. and M.C. J.L.F., C.W. and A.D. analysed the data. All authors helped to draft the manuscript and read and approved the final manuscript.

Competing interests. We declare we have no competing interests.
Funding. Research was supported by the Manchester University NHS Foundation Trust, the University of Manchester, the National Institutes of Health Research, a Knut and Alice Wallenberg Academy Fellowship (2016-0146) and Swedish Research Council Grant (2017-04680) to J.L.F., and a Wenner Gren Postdoctoral Fellowship to A.D.

Acknowledgements. We thank the patients who consented to their sperm and follicular fluid being used in this research and the staff at the Department of Reproductive Medicine, St Mary's Hospital, Manchester for making this study possible, particularly Claudette Wright and Chelsea Buck. We also thank Andrea Pilastro for use of his sperm tracker.

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
