## [Reviewer comments · Proceedings of the Royal Society B: Biological Sciences]

Review History

RSPB-2019-2262.R0 (Original submission)

Review form: Reviewer 1

Recommendation

Accept with minor revision (please list in comments)

Scientific importance: Is the manuscript an original and important contribution to its field?

Excellent

General interest: Is the paper of sufficient general interest?

Excellent

Quality of the paper: Is the overall quality of the paper suitable?

Excellent

Is the length of the paper justified?

Yes

Should the paper be seen by a specialist statistical reviewer?

No

Do you have any concerns about statistical analyses in this paper? If so, please specify them explicitly in your report.

No

It is a condition of publication that authors make their supporting data, code and materials available - either as supplementary material or hosted in an external repository. Please rate, if applicable, the supporting data on the following criteria.

Is it accessible?

No

Is it clear?

N/A

Is it adequate?

N/A

Do you have any ethical concerns with this paper?

No

Comments to the Author

General comments:

This manuscript describes interactions between female ovarian fluids and male sperm in humans, using a combination of simultaneous and non-simultaneous postmating choice experiments between gametes of real-life partners and non-partners, and microscopic measures of sperm performance in different females and a control medium. The authors show that the identity of the male-female combination used matters in terms of how much sperm is indeed attracted by follicular fluid, with potential important consequences for fertilization success and implications for studies of the evolution of cryptic female choice across taxa and potential applications for fertility treatments. The study is very timely, the experimental design and statistics of high quality and the discussion absolutely excellent.

My only two major comments are that I would want to see some re-ordering of some sections of the methods for clarity (see details below), and that the introduction could be somewhat fleshed out. While the discussion does a really good job at placing the study in context and spelling out the importance and potential consequences of the results presented here, the introduction is rather choice and lacks some justification of the rationale behind studying cryptic female choice in humans (including some more information on what has been found in other internal fertilizers so far) and the potential consequences of such mechanisms for fertility. I mainly worry that the jump between inter-specific gamete choice in broadcast spawners and chemical interactions between human sperm and eggs is a bit abrupt, and would want to see a better transition between these questions.

Additionally, two questions occurred to me when reading the discussion.

- 1) Would you expect a different outcome in your comparison of accumulation in partner vs non-partner follicular fluid if you were studying fertile couples that have already successfully naturally reproduced?
- 2) Do you have any subsequent measure of successful embryonal development available for these couples? If as you say, fertilization success might essentially be by-passed by the fact that methods are optimized to make fertilization happen, and if there might be negative consequences of having sperm that might not have naturally be selected actually fertilizing the eggs, could this result in differences in successful rates of subsequent embryonal development that could be linked to your measures of differential accumulation?

Line specific comments:

Line 22-23: There is a bit of an abrupt jump between the general first sentence about female-sperm interactions and going directly into egg chemoattractants specifically. Consider adding one sentence here that makes it clear that cryptic female choice is increasingly detected, even in species with premating mate choice, but that we still don't understand much about these mechanisms at the molecular level.

Line 26-27: Explain briefly what these 2 experimental conditions are. Consider splitting this new sentence so that you can explain what follicular fluid is in the first one and avoid the current, slightly awkward place of commas in this long sentence.

Line 35: It might be good to mention MHC in the abstract if it is chosen as one of the keywords

Lines 55-68: This paragraph and the introduction in general could use some more specific mentions of other mechanisms of cryptic female choice in internal fertilizers (including mice), that might or might not be mediated by chemoattractants. While I see that 2 such studies are cited in line 67, actually spelling out what was found there, and adding your current citation 29 to that paragraph could help justifying your current study, possibly avoiding a big jump between internal fertilizers using chemoattractants for species recognitions, mammals lacking species recognition on the sperm surface, and the interest of looking into chemoattractants and postmating mate choice. While this might not have been explored in humans at all, others have found differential fertilization success between males of the same species mediated by interactions with some part of the female reproductive tract, including in mammals, and it is thus perfectly logical to examine these mechanisms in human.

Line 55: It is slightly unclear what is meant by "a sexually selected function", please develop.

Line 74: add a few words about how sperm behaviour is altered when in contact with follicular fluid.

Line 78: briefly describe the two distinct experimental designs here already.

Line 91: I think that the differences between treatments, potential consequences on sperm behaviour and what was done to take these into account need to be explained here already and suggest changing the order of the paragraphs to have "clinical data" right after "sample acquisition". I wondered about if you considered this at all for the whole rest of the methods and would have appreciated reading this earlier.

Line 111: add the values for the repeatability in the main text

Line 112: Explain what you mean by chemotactic response rather than trapping effect.

Line 117: 2 against 14 ICSI patients seems highly unbalanced. You write later that removing them didn't change the results, I would add this information here as well.

Line 170: this statement is misleading as this proportion is highly different in the 2 experiments (i.e. 1/8 vs 1/2 of patients), this should be described as it is.

Line 184: investigate, not "investigated"

Line 199: assess, not "assessed", same line 228

Line 331: remove "suggests"

Line 350: fertilizing instead of "fertilize"

Line 384: influencing instead of “influences”

Line 414: The sentence is missing a word.

Line 521: You may want to change the wording to fertility treatment instead of in vitro fertilization as this is used to describe one of two types of fertility treatments throughout the manuscript.

Supplementary Information:

line 28: you write “although subsequent data analyses differentiated between IVF and ICSI treatments (described below)”, this is in the main text and not “below” as far as I can tell (apart from figure S2 legend).

Line 87: behaviour

Line 104: responses instead of responses

Line 151: “in” follicular fluid

Review form: Reviewer 2

Recommendation

Major revision is needed (please make suggestions in comments)

Scientific importance: Is the manuscript an original and important contribution to its field?

Excellent

General interest: Is the paper of sufficient general interest?

Excellent

Quality of the paper: Is the overall quality of the paper suitable?

Good

Is the length of the paper justified?

Yes

Should the paper be seen by a specialist statistical reviewer?

Yes

Do you have any concerns about statistical analyses in this paper? If so, please specify them explicitly in your report.

Yes

It is a condition of publication that authors make their supporting data, code and materials available - either as supplementary material or hosted in an external repository. Please rate, if applicable, the supporting data on the following criteria.

Is it accessible?

N/A

Is it clear?

N/A

Is it adequate?

N/A

Do you have any ethical concerns with this paper?

No

Comments to the Author

General comments:

This paper reports in-vitro experiments in humans, designed to test the influence of female ID, male ID and their interaction on sperm attraction to follicular fluid. Using two different experimental designs (simultaneous/non-simultaneous exposure of sperm to the follicular fluid of two females (partner and non-partner)), the authors find significant interaction effects in both experiments (as well as significant (but ignored – see below) effects of Male ID in one and Female ID in the other experiment), but no preference for the fluid of the partner, no effects on sperm swimming speed and little evidence for effects on fertilization success.

Performing such sperm/follicular fluid experiments on humans is quite exceptional and I commend the authors for being able to get access to samples, getting participant consent and permits sorted and performing these logistically rather demanding experiments. The results are of general interest, both scientifically and due to the potential implications for human infertility issues. My main reservations/questions relate to the statistics and the lack of discussion of significant effects of male ID and female ID. First, it is confusing how the terms “sperm accumulation” and “sperm responsiveness” are used in the Statistics chapter. In the ANOVAs, these terms are used for response variables in simultaneous and non-simultaneous choice experiments, respectively. In the GLMMs, “sperm accumulation” is used for both experiments. Finally, in the tests of fertilization success, “sperm responsiveness” is used for both experiments. The paper would benefit from a clarification of these terms and their use. Second, regarding the GLMMs. I have problems understanding how sperm accumulation (number of sperm accumulating; line 190) in either partner or non-partner can be a binomial response variable. I would perhaps understand it if this related to the “choice” of single sperm (although in that case there would actually be three outcomes, as the sperm could end up in one of the two follicular fluid capillaries or outside the capillaries). In any case, I don’t think this analysis is done on single sperm. Perhaps sperm accumulation is scored as either 0 (or some other categorical value): majority of sperm accumulating in partner or 1: majority of sperm accumulating in non-partner, but this is far from clear from the text. The number of random factors (5?) also seem overly high for the sample size in these models. In sum, the GLMMs need to be explained in greater detail. Third, why are significant male ID effects in the ANOVA from the simultaneous choice experiment, and significant female ID effects in the ANOVA from the non-simultaneous choice experiment (apparent in Table 1 and Figure 2), not mentioned in the Result text and the Discussion? In particular, the effect of female ID in the latter experiment is highly significant. Does not this in fact suggest an additional effect of chemoattractant amount or quality, opposite to the statement on page 12, lines 329-330? I see there is a note on the male effect in the supplementary (page 3, line 62-64), but these effects need to be addressed in the main text.

Specific comments:

Title: the title is somewhat un-descriptive of the content of the paper, as it does not convey the point about the interaction effect, and none of the experiments involved sperm from different males but rather eggs from different females.

Keywords. The keyword “MHC” should be deleted, as this paper has no data on MHC.

Page 2, line 21-22. “...chemical communication between females and sperm” seems a bit odd. I suggest changing to “...between the female reproductive system and sperm”.

Page 4, lines 74-76. This sentence is strange. First, how can sperm behavioural responses differ among women? Second, what is meant by that sperm behavioural responses to follicular fluid are

associated with increased fertilization rates? Please rewrite.

Page 4, line 93 and 96. The same typo twice: ensure, not ensured. Similar duplicated typo in line 199 (page 8) and 228 (page 9): assess, not assessed.

Page 12, line 331. Delete "suggests".

Page 13. Lines 366-385. The discussion about possible mechanisms is interesting, but it strikes me that one possibility is not really discussed, namely that in cases where the sperm do not preferentially swim towards the partner's follicular fluid, this could be due to lack of genetic compatibility between partners (or at least better compatibility with the non-partner), which could be (part of) the reason these couples struggle to have children in the first place. In other words, not pathology, just incompatibility. This might be a sensitive issue for the participants, of course, but it seems strange not to mention this possibility in the paper.

Page 13, lines 373-374. The selection of papers supporting/not supporting MHC-dependent mate choice in humans seem biased towards the latter, with only a very old reference (albeit one of the original ones) in favour of the former. There are several more recent papers supporting this, e.g. Chaix, R., C. Cao and P. Donnelly (2008). "Is mate choice in humans MHC-dependent?" *PLoS genetics* 4(9): e1000184-e1000184.

Winternitz, J., J. L. Abbate, E. Huchard, J. Havlíček and L. Z. Garamszegi (2017). "Patterns of MHC-dependent mate selection in humans and nonhuman primates: a meta-analysis." *Molecular Ecology* 26(2): 668-688.

Supplementary page 2, line 51. Please add the sample size for how many samples that was possible to standardize.

Supplementary page 4, 104. Change to "responses"

Decision letter (RSPB-2019-2262.R0)

04-Nov-2019

Dear Dr Fitzpatrick:

I am writing to inform you that your manuscript RSPB-2019-2262 entitled "Chemical signals from eggs preferentially attracts sperm from different males in humans" has, in its current form, been rejected for publication in *Proceedings B*.

This action has been taken on the advice of referees, who have recommended that substantial revisions are necessary. With this in mind we would be happy to consider a resubmission, provided the comments of the referees are fully addressed. However please note that this is not a provisional acceptance.

Please find below the comments made by the referees, not including confidential reports to the

Editor, which I hope you will find useful. If you do choose to resubmit your manuscript, please upload the following:

Sincerely,
 Professor Hans Heesterbeek
 mailto: proceedingsb@royalsociety.org

Associate Editor

Comments to Author:

We have now obtained two expert and thoughtful reviews of your manuscript. We all agree that the study is timely, well written and presents important results that will be of interest to a broad readership. Here, the authors conduct a number of logistically demanding in-vitro experiments in humans to test female-male interactions in internal fertilisation. They find that responses in human sperm behaviour to chemical signals released from human eggs depends on the specific identities of females and males, with important consequences for understanding cryptic female mate choice, sperm competition, and fertility more broadly. However, both reviewers raised important points that need to be addressed before this manuscript can be published.

Most notably:

Reviewer 1 had a number of suggestions for improving the clarity and readability of the manuscript. In particular, their main concern was a lack of detail in the Introduction. They have a number of thoughtful suggestions for how to improve this.

Reviewer 2 highlighted a number of statistical concerns. In particular, they raised a number of questions about the definition of the response variables used in GLMMs for simultaneous and non-simultaneous choice experiments. This should be dealt with fully. Finally, they point out that there are significant effects of Male ID in the simultaneous choice experiment and Female ID in the non-simultaneous choice experiment but these are not discussed.

Finally, both reviewers point out that the data is not yet publically available. This should be remedied before publication.

Reviewer(s)' Comments to Author:

Referee: 1

Comments to the Author(s)

General comments:

This manuscript describes interactions between female ovarian fluids and male sperm in humans, using a combination of simultaneous and non-simultaneous postmating choice experiments between gametes of real-life partners and non-partners, and microscopic measures of sperm performance in different females and a control medium. The authors show that the identity of the

male-female combination used matters in terms of how much sperm is indeed attracted by follicular fluid, with potential important consequences for fertilization success and implications for studies of the evolution of cryptic female choice across taxa and potential applications for fertility treatments. The study is very timely, the experimental design and statistics of high quality and the discussion absolutely excellent.

My only two major comments are that I would want to see some re-ordering of some sections of the methods for clarity (see details below), and that the introduction could be somewhat fleshed out. While the discussion does a really good job at placing the study in context and spelling out the importance and potential consequences of the results presented here, the introduction is rather choice and lacks some justification of the rationale behind studying cryptic female choice in humans (including some more information on what has been found in other internal fertilizers so far) and the potential consequences of such mechanisms for fertility. I mainly worry that the jump between inter-specific gamete choice in broadcast spawners and chemical interactions between human sperm and eggs is a bit abrupt, and would want to see a better transition between these questions.

Additionally, two questions occurred to me when reading the discussion.

1) Would you expect a different outcome in your comparison of accumulation in partner vs non-partner follicular fluid if you were studying fertile couples that have already successfully naturally reproduced?

2) Do you have any subsequent measure of successful embryonal development available for these couples? If as you say, fertilization success might essentially be by-passed by the fact that methods are optimized to make fertilization happen, and if there might be negative consequences of having sperm that might not have naturally be selected actually fertilizing the eggs, could this result in differences in successful rates of subsequent embryonal development that could be linked to your measures of differential accumulation?

Line specific comments:

Line 22-23: There is a bit of an abrupt jump between the general first sentence about female-sperm interactions and going directly into egg chemoattractants specifically. Consider adding one sentence here that makes it clear that cryptic female choice is increasingly detected, even in species with premating mate choice, but that we still don't understand much about these mechanisms at the molecular level.

Line 26-27: Explain briefly what these 2 experimental conditions are. Consider splitting this new sentence so that you can explain what follicular fluid is in the first one and avoid the current, slightly awkward place of commas in this long sentence.

Line 35: It might be good to mention MHC in the abstract if it is chosen as one of the keywords

Lines 55-68: This paragraph and the introduction in general could use some more specific mentions of other mechanisms of cryptic female choice in internal fertilizers (including mice), that might or might not be mediated by chemoattractants. While I see that 2 such studies are cited in line 67, actually spelling out what was found there, and adding your current citation 29 to that paragraph could help justifying your current study, possibly avoiding a big jump between internal fertilizers using chemoattractants for species recognitions, mammals lacking species recognition on the sperm surface, and the interest of looking into chemoattractants and postmating mate choice. While this might not have been explored in humans at all, others have found differential fertilization success between males of the same species mediated by interactions with some part of the female reproductive tract, including in mammals, and it is thus perfectly logical to examine these mechanisms in human.

Line 55: It is slightly unclear what is meant by "a sexually selected function", please develop.

Line 74: add a few words about how sperm behaviour is altered when in contact with follicular fluid.

Line 78: briefly describe the two distinct experimental designs here already.

Line 91: I think that the differences between treatments, potential consequences on sperm behaviour and what was done to take these into account need to be explained here already and suggest changing the order of the paragraphs to have "clinical data" right after "sample acquisition". I wondered about if you considered this at all for the whole rest of the methods and would have appreciated reading this earlier.

Line 111: add the values for the repeatability in the main text

Line 112: Explain what you mean by chemotactic response rather than trapping effect.

Line 117: 2 against 14 ICSI patients seems highly unbalanced. You write later that removing them didn't change the results, I would add this information here as well.

Line 170: this statement is misleading as this proportion is highly different in the 2 experiments (i.e. 1/8 vs 1/2 of patients), this should be described as it is.

Line 184: investigate, not "investigated"

Line 199: assess, not "assessed", same line 228

Line 331: remove "suggests"

Line 350: fertilizing instead of "fertilize"

Line 384: influencing instead of "influences"

Line 414: The sentence is missing a word.

Line 521: You may want to change the wording to fertility treatment instead of in vitro fertilization as this is used to describe one of two types of fertility treatments throughout the manuscript.

Supplementary Information:

line 28: you write "although subsequent data analyses differentiated between IVF and ICSI treatments (described below)", this is in the main text and not "below" as far as I can tell (apart from figure S2 legend).

Line 87: behaviour

Line 104: responses instead of responses

Line 151: "in" follicular fluid

Referee: 2

Comments to the Author(s)

General comments:

This paper reports in-vitro experiments in humans, designed to test the influence of female ID, male ID and their interaction on sperm attraction to follicular fluid. Using two different

experimental designs (simultaneous/non-simultaneous exposure of sperm to the follicular fluid of two females (partner and non-partner)), the authors find significant interaction effects in both experiments (as well as significant (but ignored – see below) effects of Male ID in one and Female ID in the other experiment), but no preference for the fluid of the partner, no effects on sperm swimming speed and little evidence for effects on fertilization success.

Performing such sperm/follicular fluid experiments on humans is quite exceptional and I commend the authors for being able to get access to samples, getting participant consent and permits sorted and performing these logistically rather demanding experiments. The results are of general interest, both scientifically and due to the potential implications for human infertility issues. My main reservations/questions relate to the statistics and the lack of discussion of significant effects of male ID and female ID. First, it is confusing how the terms “sperm accumulation” and “sperm responsiveness” are used in the Statistics chapter. In the ANOVAs, these terms are used for response variables in simultaneous and non-simultaneous choice experiments, respectively. In the GLMMs, “sperm accumulation” is used for both experiments. Finally, in the tests of fertilization success, “sperm responsiveness” is used for both experiments. The paper would benefit from a clarification of these terms and their use. Second, regarding the GLMMs. I have problems understanding how sperm accumulation (number of sperm accumulating; line 190) in either partner or non-partner can be a binomial response variable. I would perhaps understand it if this related to the “choice” of single sperm (although in that case there would actually be three outcomes, as the sperm could end up in one of the two follicular fluid capillaries or outside the capillaries). In any case, I don’t think this analysis is done on single sperm. Perhaps sperm accumulation is scored as either 0 (or some other categorical value): majority of sperm accumulating in partner or 1: majority of sperm accumulating in non-partner, but this is far from clear from the text. The number of random factors (5?) also seem overly high for the sample size in these models. In sum, the GLMMs need to be explained in greater detail. Third, why are significant male ID effects in the ANOVA from the simultaneous choice experiment, and significant female ID effects in the ANOVA from the non-simultaneous choice experiment (apparent in Table 1 and Figure 2), not mentioned in the Result text and the Discussion? In particular, the effect of female ID in the latter experiment is highly significant. Does not this in fact suggest an additional effect of chemoattractant amount or quality, opposite to the statement on page 12, lines 329-330? I see there is a note on the male effect in the supplementary (page 3, line 62-64), but these effects need to be addressed in the main text.

Specific comments:

Title: the title is somewhat un-descriptive of the content of the paper, as it does not convey the point about the interaction effect, and none of the experiments involved sperm from different males but rather eggs from different females.

Keywords. The keyword “MHC” should be deleted, as this paper has no data on MHC.

Page 2, line 21-22. “...chemical communication between females and sperm” seems a bit odd. I suggest changing to “...between the female reproductive system and sperm”.

Page 4, lines 74-76. This sentence is strange. First, how can sperm behavioural responses differ among women? Second, what is meant by that sperm behavioural responses to follicular fluid are associated with increased fertilization rates? Please rewrite.

Page 4, line 93 and 96. The same typo twice: ensure, not ensured. Similar duplicated typo in line 199 (page 8) and 228 (page 9): assess, not assessed.

Page 12, line 331. Delete “suggests”.

Page 13. Lines 366-385. The discussion about possible mechanisms is interesting, but it strikes me that one possibility is not really discussed, namely that in cases where the sperm do not preferentially swim towards the partner's follicular fluid, this could be due to lack of genetic compatibility between partners (or at least better compatibility with the non-partner), which

could be (part of), the reason these couples struggle to have children in the first place. In other words, not pathology, just incompatibility. This might be a sensitive issue for the participants, of course, but it seems strange not to mention this possibility in the paper.

Page 13, lines 373-374. The selection of papers supporting/not supporting MHC-dependent mate choice in humans seem biased towards the latter, with only a very old reference (albeit one of the original ones) in favour of the former. There are several more recent papers supporting this, e.g. Chaix, R., C. Cao and P. Donnelly (2008). "Is mate choice in humans MHC-dependent?" *PLoS genetics* 4(9): e1000184-e1000184.
Winternitz, J., J. L. Abbate, E. Huchard, J. Havlíček and L. Z. Garamszegi (2017). "Patterns of MHC-dependent mate selection in humans and nonhuman primates: a meta-analysis." *Molecular Ecology* 26(2): 668-688.

Supplementary page 2, line 51. Please add the sample size for how many samples that was possible to standardize.

Supplementary page 4, 104. Change to "responses"

Author's Response to Decision Letter for (RSPB-2019-2262.R0)

See Appendix A.

RSPB-2020-0805.R0

Review form: Reviewer 2

Recommendation

Accept as is

Scientific importance: Is the manuscript an original and important contribution to its field?

Excellent

General interest: Is the paper of sufficient general interest?

Excellent

Quality of the paper: Is the overall quality of the paper suitable?

Excellent

Is the length of the paper justified?

Yes

Should the paper be seen by a specialist statistical reviewer?

Yes

Do you have any concerns about statistical analyses in this paper? If so, please specify them explicitly in your report.

No

It is a condition of publication that authors make their supporting data, code and materials available - either as supplementary material or hosted in an external repository. Please rate, if applicable, the supporting data on the following criteria.

Is it accessible?

Yes

Is it clear?

Yes

Is it adequate?

No

Do you have any ethical concerns with this paper?

No

Comments to the Author

The authors have done a very nice job answering all my comments and concerns. No further comments.

Decision letter (RSPB-2020-0805.R0)

10-May-2020

Dear Dr Fitzpatrick

I am pleased to inform you that your manuscript RSPB-2020-0805 entitled "Chemical signals from eggs facilitates cryptic female choice in humans" has been accepted for publication in Proceedings B.

The referee and the Associate Editor have recommended publication, but also suggest some very minor revisions to your manuscript. Therefore, I invite you to respond to the comments and revise your manuscript. Because the schedule for publication is very tight, it is a condition of publication that you submit the revised version of your manuscript within 7 days. If you do not think you will be able to meet this date please let us know.

Sincerely,
Professor Hans Heesterbeek
mailto: proceedingsb@royalsociety.org

Associate Editor

Board Member

Comments to Author:

The authors have done a very thorough job revising the manuscript and it is much improved as a result.

As one of the reviewers highlights, the code is not publicly available. It is a condition of publication that authors make their supporting data, code and materials available - either as supplementary material or hosted in an external repository. Can the authors deposit the code for their statistical analyses.

Given that the authors discuss the MHC in relation to partner genetic compatibility and reference 4 papers regarding MHC-dependent mate selection, I think it is an appropriate keyword.

L223 missing 'the' before microcapillary tube

L274 missing '(' before 0

Reviewer(s)' Comments to Author:

Referee: 2

Comments to the Author(s).

The authors have done a very nice job answering all my comments and concerns. No further comments.

Author's Response to Decision Letter for (RSPB-2020-0805.R0)

See Appendix B.

Decision letter (RSPB-2020-0805.R1)

20-May-2020

Dear Dr Fitzpatrick

I am pleased to inform you that your manuscript entitled "Chemical signals from eggs facilitates cryptic female choice in humans" has been accepted for publication in Proceedings B.

Your article has been estimated as being 9 pages long. Our Production Office will be able to confirm the exact length at proof stage.

Open Access

Paper charges

Sincerely,

Appendix A

Dear Professor Heesterbeek and Associate Editor Board Member,

Thank you for your positive comments on the last version of our manuscript and for the opportunity to resubmit our work. We have taken on board the many useful comments offered during the last round of review and have revised our manuscript accordingly.

In the revised manuscript we now 1) clarify various aspects of the manuscript, following the reviewer suggestions, 2) address the concerns about our statistical analyses by explaining our modelling approach in more detail and adding additional analyses to the Supporting Material to bolster our main findings, and 3) include the link to the data in Dryad for the reviewers.

We feel that attending to these comments has helped strengthen our manuscript and conclusions. We hope you agree.

Best wishes,

John Fitzpatrick
(on behalf of the other authors)

04-Nov-2019

Dear Dr Fitzpatrick:

I am writing to inform you that your manuscript RSPB-2019-2262 entitled "Chemical signals from eggs preferentially attracts sperm from different males in humans" has, in its current form, been rejected for publication in Proceedings B.

This action has been taken on the advice of referees, who have recommended that substantial revisions are necessary. With this in mind we would be happy to consider a resubmission, provided the comments of the referees are fully addressed. However please note that this is not a provisional acceptance.

1) A 'response to referees' document including details of how you have responded to the comments, and the adjustments you have made.

- 2) A clean copy of the manuscript and one with 'tracked changes' indicating your 'response to referees' comments document.
- 3) Line numbers in your main document.

Sincerely,

Professor Hans Heesterbeek
mailto:proceedingsb@royalsociety.org

Associate Editor

Comments to Author:

We have now obtained two expert and thoughtful reviews of your manuscript. We all agree that the study is timely, well written and presents important results that will be of interest to a broad readership. Here, the authors conduct a number of logistically demanding in-vitro experiments in humans to test female-male interactions in internal fertilisation. They find that responses in human sperm behaviour to chemical signals released from human eggs depends on the specific identities of females and males, with important consequences for understanding cryptic female mate choice, sperm competition, and fertility more broadly. However, both reviewers raised important points that need to be addressed before this manuscript can be published.

Most notably:

Reviewer 1 had a number of suggestions for improving the clarity and readability of the manuscript. In particular, their main concern was a lack of detail in the Introduction. They have a number of thoughtful suggestions for how to improve this.

Reviewer 2 highlighted a number of statistical concerns. In particular, they raised a number of questions about the definition of the response variables used in GLMMs for simultaneous and non-simultaneous choice experiments. This should be dealt with fully. Finally, they point out that there are significant effects of Male ID in the simultaneous choice experiment and Female ID in the non-simultaneous choice experiment but these are not discussed.

Finally, both reviewers point out that the data is not yet publically available. This should be remedied before publication.

Response: We would like to thank the editor and reviewers for their enthusiastic response to our work. We have taken on board the constructive criticism and we believe that the

manuscript is now considerably improved. In addition, we have now made the data available for review in Dryad (<https://datadryad.org/stash/share/ljigszrWfHAO1HiCbbnzPvnxW6LKgz8sc-Om1F3gZIs>) and include this information in the revised manuscript.

Reviewer(s)' Comments to Author:

Referee: 1

Comments to the Author(s)

General comments:

This manuscript describes interactions between female ovarian fluids and male sperm in humans, using a combination of simultaneous and non-simultaneous postmating choice experiments between gametes of real-life partners and non-partners, and microscopic measures of sperm performance in different females and a control medium. The authors show that the identity of the male-female combination used matters in terms of how much sperm is indeed attracted by follicular fluid, with potential important consequences for fertilization success and implications for studies of the evolution of cryptic female choice across taxa and potential applications for fertility treatments. The study is very timely, the experimental design and statistics of high quality and the discussion absolutely excellent.

Response: Thank you for your generous comments about the manuscript, this is much appreciated.

My only two major comments are that I would want to see some re-ordering of some sections of the methods for clarity (see details below), and that the introduction could be somewhat fleshed out. While the discussion does a really good job at placing the study in context and spelling out the importance and potential consequences of the results presented here, the introduction is rather choice and lacks some justification of the rationale behind studying cryptic female choice in humans (including some more information on what has been found in other internal fertilizers so far) and the potential consequences of such mechanisms for fertility. I mainly worry that the jump between inter-specific gamete choice in broadcast spawners and chemical interactions between human sperm and eggs is a bit abrupt, and would want to see a better transition between these questions.

Response: We have added more detail to the Introduction as requested (lines 65-77) and agree that this added information helps smooth the transition from broadcast spawning species to mammals.

Additionally, two questions occurred to me when reading the discussion.

1) Would you expect a different outcome in your comparison of accumulation in partner vs non-partner follicular fluid if you were studying fertile couples that have already successfully naturally reproduced?

Response: This is a great question and one that we thought about as well. The simple answer is we don't know. Getting these samples from fertile couples that have already successfully naturally reproduced would be challenging and moving not a non-human model system may be the better choice for investigating this. Yet this is clearly an open question from our study and represents a natural next step to consider. Therefore, we added a sentence in the concluding paragraph stating that a critical next step is to determine if the patterns we report here are broadly applicable in mammalian reproduction, including in humans not seeking assisted fertility treatments (lines 431-435).

2) Do you have any subsequent measure of successful embryonal development available for these couples? If as you say, fertilization success might essentially be by-passed by the fact that methods are optimized to make fertilization happen, and if there might be negative consequences of having sperm that might not have naturally be selected actually fertilizing the eggs, could this result in differences in successful rates of subsequent embryonal development that could be linked to your measures of differential accumulation?

Response: This is another great point. We agree that this is a very interesting avenue for consideration and we now include data where we could address this. Standard clinical data includes embryo quality scores either at the early cleavage or blastocyst stage and we include these data where possible. In our study population, some of the couple became pregnant and of these many gave live birth. We now include this information in the revised manuscript (lines 124-127, with further information provide in the Supplementary Material). However, neither embryo quality, pregnancy nor live birth values were associated with any difference in sperm accumulation or responsiveness (lines 334-345). This provides some limited reassurance that our findings are not purely associated with degree of couple subfertility and speaks in part to the referee's first question above.

Line specific comments:

Line 22-23: There is a bit of an abrupt jump between the general first sentence about female-sperm interactions and going directly into egg chemoattractants specifically. Consider adding one sentence here that makes it clear that cryptic female choice is increasingly detected, even in species with premating mate choice, but that we still don't understand much about these mechanisms at the molecular level.

Response: Done.

Line 26-27: Explain briefly what these 2 experimental conditions are. Consider splitting this new sentence so that you can explain what follicular fluid is in the first one and avoid the current, slightly awkward place of commas in this long sentence.

Response: Done.

Line 35: It might be good to mention MHC in the abstract if it is chosen as one of the keywords

Response: Keywords are not meant overlap with words used in the title or abstract as these words are already retrievable by search engines. Instead the keywords are meant to provide additional context to a work and expose a particular manuscript to searches on related topics that would not come up when searching for the words in the title and abstract. As we did not directly address MHC in this study we do not feel it is appropriate to add it to the abstract. However, we did want to include it in the keywords to alert readers who are interested in MHC of its potential in explaining our findings.

Lines 55-68: This paragraph and the introduction in general could use some more specific mentions of other mechanisms of cryptic female choice in internal fertilizers (including mice), that might or might not be mediated by chemoattractants. While I see that 2 such studies are cited in line 67, actually spelling out what was found there, and adding your current citation 29 to that paragraph could help justifying your current study, possibly avoiding a big jump between internal fertilizers using chemoattractants for species recognitions, mammals lacking species recognition on the sperm surface, and the interest of looking into chemoattractants and postmating mate choice. While this might not have been explored in humans at all, others have found differential fertilization success between males of the same species mediated by interactions with some part of the female reproductive tract, including in mammals, and it is thus perfectly logical to examine these mechanisms in human.

Response: Great point. We have added more detail about cryptic female choice generally in this paragraph and added information highlighting the potential for chemoattractants to be used as mechanism for facilitating cryptic female choice in non-human mammalian models (lines 56-82). We agree with the Referee that this addition makes the jump between marine invertebrates and mammals less abrupt and we thank them for the suggestion.

Line 55: It is slightly unclear what is meant by “a sexually selected function”, please develop.

Response: To clarify we re-wrote this sentence to specifically state that chemoattractants can facilitate sperm and eggs to exercise gamete-mediated mate choice (lines 61-62).

Line 74: add a few words about how sperm behaviour is altered when in contact with follicular fluid.

Response: Done (lines 88-89).

Line 78: briefly describe the two distinct experimental designs here already.

Response: Done (lines 95-96).

Line 91: I think that the differences between treatments, potential consequences on sperm

behaviour and what was done to take these into account need to be explained here already and suggest changing the order of the paragraphs to have “clinical data” right after “sample acquisition”. I wondered about if you considered this at all for the whole rest of the methods and would have appreciated reading this earlier.

Response: Good idea. We have now restructured the methods accordingly.

Line 111: add the values for the repeatability in the main text

Response: Done (lines 152-153).

Line 112: Explain what you mean by chemotactic response rather than trapping effect.

Response: Done (lines 153-156).

Line 117: 2 against 14 ICSI patients seems highly unbalanced. You write later that removing them didn't change the results, I would add this information here as well.

Response: Done (lines 161-162).

Line 170: this statement is misleading as this proportion is highly different in the 2 experiments (i.e. 1/8 vs 1/2 of patients), this should be described as it is.

Response: We have now removed this statement. In restructuring the methods section in line with the Referee's suggestions it no longer made sense to draw this distinction between IVF and ICSI patients at this point in the methods. Instead we describe how these two groups differ and leave the description of how these two groups made up the samples of the simultaneous and non-simultaneous experiments for when these experiments are described in full later in the methods.

Line 184: investigate, not “investigated”

Response: Done.

Line 199: assess, not “assessed”, same line 228

Response: Done and done.

Line 331: remove “suggests”

Response: Done.

Line 350: fertilizing instead of “fertilize”

Response: Done.

Line 384: influencing instead of “influences”

Response: Done.

Line 414: The sentence is missing a word.

Response: Corrected.

Line 521: You may want to change the wording to fertility treatment instead of in vitro fertilization as this is used to describe one of two types of fertility treatments throughout the manuscript.

Response: Done. We now use the term ‘clinical assisted reproductive treatment’ throughout the manuscript.

Supplementary Information:

line 28: you write “although subsequent data analyses differentiated between IVF and ICSI treatments (described below)”, this is in the main text and not “below” as far as I can tell (apart from figure S2 legend).

Response: Corrected.

Line 87: behaviour

Response: Done.

Line 104: responses instead of responses

Response: Done.

Line 151: “in” follicular fluid

Response: Done.

Referee: 2

Comments to the Author(s)

General comments:

This paper reports in-vitro experiments in humans, designed to test the influence of female ID, male ID and their interaction on sperm attraction to follicular fluid. Using two different experimental designs (simultaneous/non-simultaneous exposure of sperm to the follicular fluid

of two females (partner and non-partner)), the authors find significant interaction effects in both experiments (as well as significant (but ignored – see below) effects of Male ID in one and Female ID in the other experiment), but no preference for the fluid of the partner, no effects on sperm swimming speed and little evidence for effects on fertilization success.

Performing such sperm/follicular fluid experiments on humans is quite exceptional and I commend the authors for being able to get access to samples, getting participant consent and permits sorted and performing these logistically rather demanding experiments. The results are of general interest, both scientifically and due to the potential implications for human infertility issues.

Response: We thank the referee for their kind words.

My main reservations/questions relate to the statistics and the lack of discussion of significant effects of male ID and female ID.

Response: We deal with this point below when the reviewer expands on their reservation.

First, it is confusing how the terms “sperm accumulation” and “sperm responsiveness” are used in the Statistics chapter. In the ANOVAs, these terms are used for response variables in simultaneous and non-simultaneous choice experiments, respectively. In the GLMMs, “sperm accumulation” is used for both experiments. Finally, in the tests of fertilization success, “sperm responsiveness” is used for both experiments. The paper would benefit from a clarification of these terms and their use.

Response: This is a great point. We now take care to clearly define all the terms used in the analyses and we use these terms consistently throughout the methods and results. For clarity we now specify four response variable terms, including:

- 1. Sperm accumulation:** the number of sperm counted in the microcapillary tube, which was the value used in analyses in the simultaneous choice experiment.
- 2. Sperm responsiveness:** the difference in the number of sperm in the follicular fluid (FF) microcapillary tube vs. the number of sperm in the control microcapillary tube (i.e. sperm number in FF – sperm number in control) in the non-simultaneous choice experiment.
- 3. Partner sperm preference:** the difference in sperm accumulation in partner vs. non-partner follicular fluid in the simultaneous choice experiment.
- 4. Partner sperm responsiveness:** the difference in the number of sperm in the follicular fluid microcapillary tube of the partner vs. the control microcapillary tube in the non-simultaneous choice experiment.

These terms are now clearly defined in the text (lines 215-217, 221-223, 275-278, and 278-280).

Second, regarding the GLMMs. I have problems understanding how sperm accumulation (number of sperm accumulating; line 190) in either partner or non-partner can be a binomial

response variable. I would perhaps understand it if this related to the “choice” of single sperm (although in that case there would actually be three outcomes, as the sperm could end up in one of the two follicular fluid capillaries or outside the capillaries). In any case, I don’t think this analysis is done on single sperm. Perhaps sperm accumulation is scored as either 0 (or some other categorical value): majority of sperm accumulating in partner or 1: majority of sperm accumulating in non-partner, but this is far from clear from the text. The number of random factors (5?) also seem overly high for the sample size in these models. In sum, the GLMMs need to be explained in greater detail.

Response: Sperm accumulation is treated as a binomial response variable because the ‘choice’ of a single sperm to accumulate in one female’s follicular fluid necessarily influences the number of sperm that can accumulate in the other female’s follicular fluid. If more sperm ‘choose’ to accumulate in the follicular fluid of female 1 then there are less available sperm left to ‘choose’ to accumulate in the follicular fluid of female 2. Sperm choice therefore is paired between the two microcapillary tubes in the simultaneous choice experiment and that is accounted for in the binary nature of our analysis. In effect all sperm added to the petri dish had the binary choice of swimming towards one follicular fluid sample or the other (see Figure 1c for a visual representation of this choice). The reviewer is correct that this analysis isn’t done on a single sperm, and we now update Figure 1 to show multiple sperm in our diagram to avoid readers getting the wrong idea. Also, the reviewer is correct that there is a third option of sperm being outside the capillaries, but since we were interested in which sperm made a ‘choice’ and treat sperm outside the capillaries as not having made a choice at the time of sampling. We have added more detail to the statistical section to make this clearer to the reader (lines 231-248).

Finally, we also explored how the number of random factors in our models influence the model results by performed a series of analyses with a reduced number of random effects. In all cases the results remained the same. This is now detailed in the Supporting Material under the ‘Additional statistical analyses: sperm accumulation and responsiveness models’ section.

Third, why are significant male ID effects in the ANOVA from the simultaneous choice experiment, and significant female ID effects in the ANOVA from the non-simultaneous choice experiment (apparent in Table 1 and Figure 2), not mentioned in the Result text and the Discussion? In particular, the effect of female ID in the latter experiment is highly significant. Does not this in fact suggest an additional effect of chemoattractant amount or quality, opposite to the statement on page 12, lines 329-330? I see there is a note on the male effect in the supplementary (page 3, line 62-64), but these effects need to be addressed in the main text.

Response: We did not interpret the main effects from our models as they occurred in the presence of a significant male-by-female interaction effect. Therefore, the female (or male) main effect is influenced by a higher order interaction term. Had the interaction term not been significant we would certainly have interpreted a significant female effect in line with the referee’s suggestion – namely that chemoattractant amount or quality influences sperm

accumulation. However, due to the significant interaction term we can only say that sperm accumulation is influenced by combinatory effects of male-female identities. To clarify why significant main effects were not discussed we have now added the motivation for our decision for both males and females to the main text (lines 225-227). And to avoid confusion and unnecessarily direct focus away from our main result – the interactive effect between females and males – we have removed the statement about chemoattractant amount or quality highlighted by the referee from the revised manuscript.

Specific comments:

Title: the title is somewhat un-descriptive of the content of the paper, as it does not convey the point about the interaction effect, and none of the experiments involved sperm from different males but rather eggs from different females.

Response: We have changed the title to: Chemical signals from eggs facilitates cryptic female choice in humans. As we note in the manuscript, female-by-male interactions are a hallmark of cryptic female choice so we feel this revised title better conveys the main findings.

Keywords. The keyword “MHC” should be deleted, as this paper has no data on MHC.

Response: As with Referee 1, we have a difference in opinion as to how keywords should be used. As detailed above, keywords are used to draw in a wider audience to the work, even if the specific work doesn't deal directly with the topic of interest to that audience (i.e. those interested in MHC). Here, since we invoke MHC as a potential explanation for our findings it seemed relevant to us to include MHC as a keyword, even if we did not have any data on MHC in this manuscript. Therefore, we have elected to keep MHC in the keywords. However, given the concerns of both the referees we are happy to be guided by the editor on how to deal with this issue.

Page 2, line 21-22. “...chemical communication between females and sperm” seems a bit odd. I suggest changing to “...between the female reproductive system and sperm”.

Response: Done.

Page 4, lines 74-76. This sentence is strange. First, how can sperm behavioural responses differ among women? Second, what is meant by that sperm behavioural responses to follicular fluid are associated with increased fertilization rates? Please rewrite.

Response: Done.

Page 4, line 93 and 96. The same typo twice: ensure, not ensured. Similar duplicated typo in line 199 (page 8) and 228 (page 9): assess, not assessed.

Response: Thanks! Corrected.

Page 12, line 331. Delete “suggests”.

Response: Done.

Page 13. Lines 366-385. The discussion about possible mechanisms is interesting, but it strikes me that one possibility is not really discussed, namely that in cases where the sperm do not preferentially swim towards the partner's follicular fluid, this could be due to lack of genetic compatibility between partners (or at least better compatibility with the non-partner), which could be (part of), the reason these couples struggle to have children in the first place. In other words, not pathology, just incompatibility. This might be a sensitive issue for the participants, of course, but it seems strange not to mention this possibility in the paper.

Response: We have now expanded on this point in the Discussion (lines 402-404).

Page 13, lines 373-374. The selection of papers supporting/not supporting MHC-dependent mate choice in humans seem biased towards the latter, with only a very old reference (albeit one of the original ones) in favour of the former. There are several more recent papers supporting this, e.g. Chaix, R., C. Cao and P. Donnelly (2008). "Is mate choice in humans MHC-dependent?" *PLoS genetics* 4(9): e1000184-e1000184.

Winternitz, J., J. L. Abbate, E. Huchard, J. Havlíček and L. Z. Garamszegi (2017). "Patterns of MHC-dependent mate selection in humans and nonhuman primates: a meta-analysis." *Molecular Ecology* 26(2): 668-688.

Response: These additional references have been added.

Supplementary page 2, line 51. Please add the sample size for how many samples that was possible to standardize.

Response: Done. We have rewritten this section extensively to explain why it was not possible to standardize some of the samples and now provide details on the average (and range) of densities used in our experiments.

Supplementary page 4, 104. Change to “responses”

Response: Done.

Journal Name: Proceedings of the Royal Society B

Journal Code: RSPB

Print ISSN: 0962-8452

Online ISSN: 1471-2954

Journal Admin Email: proceedingsb@royalsociety.org

MS Reference Number: RSPB-2019-2262

Article Status: REJECTED

MS Dryad ID: RSPB-2019-2262

MS Title: Chemical signals from eggs preferentially attracts sperm from different males in humans

MS Authors: Fitzpatrick, John L; Willis, Charlotte; Devigili, Alessandro; Young, Amy; Carroll, Michael; Hunter, Helen; Brison, Daniel

Contact Author: John L Fitzpatrick

Contact Author Email: john.fitzpatrick@zoologi.su.se

Contact Author Address 1: Svante Arrhenius väg 18b

Contact Author Address 2:

Contact Author Address 3:

Contact Author City: Stockholm

Contact Author State:

Contact Author Country: Sweden

Contact Author ZIP/Postal Code: 106 91

Keywords: sperm chemotaxis, post-copulatory sexual selection, MHC, sperm competition, in vitro fertilization

Abstract: Mate choice can continue after mating via chemical communication between females and sperm. Yet, whether chemical signals released from eggs (chemoattractants) allow females to exert cryptic female choice to favour sperm from specific males remains an open question, particularly in species (including humans) where adults exercise pre-mating mate choice. Here, we adapt a classic dichotomous mate choice assay to the microscopic scale to assess gamete-mediated mate choice in humans. We report robust evidence under two distinct experimental conditions that follicular fluid, a source of human sperm chemoattractants, from different females consistently and differentially attracts sperm from specific males. This chemoattractant-moderated choice of sperm offers eggs an avenue to exercise independent mate preference. Indeed, gamete-mediated mate choice did not reinforce pre-mating human mate choice decisions. Our results demonstrate that chemoattractants facilitate gamete-mediated mate choice in humans, which offers females the opportunity to exert cryptic female choice for sperm from specific males.

EndDryadContent

Appendix B

Dear Professor Heesterbeek and Associate Editor Board Member,

Thank you for accepting our manuscript! We have now made the minor changes requested, including uploading the code for our analyses to Dryad. We look forward to seeing our work in print!

Best wishes,

John Fitzpatrick
(on behalf of the other authors)

Associate Editor
Board Member

Comments to Author:

The authors have done a very thorough job revising the manuscript and it is much improved as a result.

As one of the reviewers highlights, the code is not publicly available. It is a condition of publication that authors make their supporting data, code and materials available - either as supplementary material or hosted in an external repository. Can the authors deposit the code for their statistical analyses.

Responses: We have now added the code, along with the raw data, to Dryad.

Given that the authors discuss the MHC in relation to partner genetic compatibility and reference 4 papers regarding MHC-dependent mate selection, I think it is an appropriate keyword.

Responses: Thank you.

L223 missing 'the' before microcapillary tube

Responses: Corrected.

L274 missing '(' before 0

Responses: Corrected.

Reviewer(s)' Comments to Author:

Referee: 2

Comments to the Author(s).

The authors have done a very nice job answering all my comments and concerns. No further comments.

Responses: Thank you!